# Halochromic Silk Fabric as a Reversible pH-Sensor Based on a Novel 2-Aminoimidazole Azo Dye

**DOI:** 10.3390/polym15071730

**Published:** 2023-03-30

**Authors:** Ana Isabel Ribeiro, Bárbara Vieira, Cátia Alves, Bárbara Silva, Eugénia Pinto, Fátima Cerqueira, Renata Silva, Fernando Remião, Vasyl Shvalya, Uros Cvelbar, Jorge Padrão, Alice Maria Dias, Andrea Zille

**Affiliations:** 1Centre for Textile Science and Technology (2C2T), Department of Textile Engineering, University of Minho, Campus of Azurém, 4800-058 Guimarães, Portugal; afr@2c2t.uminho.pt (A.I.R.); barbaravieira98.bv@gmail.com (B.V.); catia.alves.98@hotmail.com (C.A.); padraoj@2c2t.uminho.pt (J.P.); 2Laboratory of Microbiology, Biological Sciences Department, Faculty of Pharmacy, University of Porto, 4050-313 Porto, Portugal; barbarapolerisilva@gmail.com (B.S.); epinto@ff.up.pt (E.P.); 3Laboratory of Toxicology, UCIBIO—Applied Molecular Biosciences Unit, REQUIMTE, Department of Biological Sciences, Faculty of Pharmacy, University of Porto, Rua de Jorge Viterbo Ferreira nº 228, 4050-313 Porto, Portugal; rsilva@ff.up.pt (R.S.); remiao@ff.up.pt (F.R.); 4Associate Laboratory i4HB—Institute for Health and Bioeconomy, Faculty of Pharmacy, University of Porto, 4050-313 Porto, Portugal; 5CIIMAR/CIMAR, Interdisciplinary Centre of Marine and Environmental Research, Terminal de Cruzeiros do Porto de Leixões, 4450-208 Matosinhos, Portugal; 6Molecular Oncology and Viral Pathology Group, Research Center of IPO Porto (CI-IPOP)/RISE@CI-IPOP (Health Research Network), Portuguese Oncology Institute of Porto (IPO Porto)/Porto Comprehensive Cancer Center (Porto. CCC), 4200-072 Porto, Portugal; fatimaf@ufp.edu.pt; 7Faculty of Health Sciences, Fernando Pessoa University, 4200-150 Porto, Portugal; 8FP-I3ID, FP-BHS, Universidade Fernando Pessoa, Praça 9 de Abril, 349, 4249-004 Porto, Portugal; 9Department of Gaseous Electronics (F6), Jožef Stefan Institute, SI-1000 Ljubljana, Slovenia; vasyl.shvalya@ijs.si (V.S.); uros.cvelbar@ijs.si (U.C.); 10Chemistry Centre of University of Minho (CQUM), Department of Chemistry, University of Minho, Campus of Gualtar, 4710-057 Braga, Portugal; ad@quimica.uminho.pt

**Keywords:** azo dyes, imidazole, silk, halochromism, smart materials, pH-responsive

## Abstract

Textiles are important components for the development of lightweight and flexible displays useful in smart materials. In particular, halochromic textiles are fibrous materials with a color-changing ability triggered by pH variations mainly based on pH-sensitive dye molecules. Recently, a novel class of 2-aminoimidazole azo dyes was developed with distinct substituent patterns. In this work, silk fabric was functionalized through exhaustion for the first time with one of these dyes (AzoIz.Pip). The halochromic properties of the dye were assessed in an aqueous solution and after silk functionalization. The solutions and the fabrics were thoroughly analyzed by ultraviolet-visible (UV-vis) spectra, color strength (K/S), color difference (∆E), CIE L*a*b* coordinates, and the ultraviolet protection factor (UPF). The dyeing process was optimized, and the halochromic performance (and reversibility) was assessed in universal Britton–Robinson buffers (ranging from pH 3 to 12) and artificial body fluids (acid and alkaline perspiration, and wound exudate). AzoIz.Pip showed vibrant colors and attractive halochromic properties with a hypsochromic shift from blue (557 nm) to magenta (536 nm) in aqueous buffered solutions. Similarly, the functionalized silk showed a shift in wavelength of the maximum K/S value from 590 nm to 560 nm when pH increases. The silk fabric showed a high affinity to AzoIz.Pip, and promoted additional color stabilization of the dye, avoiding color loss as observed when the dye is in solution at alkaline pH after 24 h. The color reversibility was effective up to the fourth cycle and the fastness tests denoted suitable results, except washing fastness. The cytotoxicity of the silk fabric extracts was assessed, depicting reduced viability of HaCaT cells to <70% only when the dye concentration in the fabric is higher or equal to 64 μg·mL^−1^. Nevertheless, lower concentrations were also very effective for the halochromic performance in silk. These materials can thus be a helpful tool for developing sensors in several sectors such as biomedicine, packaging, filtration, agriculture, protective apparel, sports, camouflage, architecture, and design.

## 1. Introduction

Textiles are important components in the development of smart materials able to sense and react to the surrounding environment, supporting people in everyday life and technical activities [1]. These textiles can be divided into passive or active and both of them change their physicochemical properties due to environmental stimuli (e.g., temperature, light, heartbeat, pH, breathing frequency, moisture, radiation, electric or magnetic field, mechanical force, sound, chemical vapors, and ions). While passive textiles only act as sensors, active textiles can act further as sensors and actuators [2,3]. Color is one of the most powerful tools of communication as a change in color can be easily perceivable and interpreted [4]. A particular group of smart materials based on substrate color is halochromic textiles, i.e., fibrous materials with a color-changing phenomenon promoted by pH variations [5]. Active textiles are used as flexible displays in several applications such as biomedical tools, packaging, filtration, agriculture, personal protective equipment, sports, camouflage, architecture, and design. This versatility is due to their simplicity and lightweight, mechanical stability, breathability, variable contact surface, and washable properties [5,6,7].

In the biomedical and fitness field, several applications of halochromic textiles can be found to unobtrusively measure specific physiological parameters in real time. Moreover, as a consequence of the increasing elderly population, home-care programs that make periodic checks more pleasant and timely, avoiding clinical interventions, are needed. Thus, considerable investments have been made to provide innovative wearable devices and technologies able to monitor health conditions and/or environmental parameters resulting in the emergence of an important market segment [8]. One example of halochromic textile applications in the healthcare sector is monitoring patients with burns or chronic wounds because the pH of the skin and exudates is a strong biomarker of the healing process. Thus, as these materials are able to indicate the healing state without the need to remove the dressing, their use may avoid unnecessary pain or discomfort to the patients [9]. Electrochemical sensors have been used within the same scope to measure pH in wounds. However, these sensors displayed limiting issues such as pain, discomfort, and lack of reliability due to the interference of biomolecules and salts, and it is also challenging to adequately sterilize them [10]. Monitoring the pH of sweat (ii) is another application that may indicate disorders prompted by the body’s hydration level, cystic fibrosis, dermatitis, bacterial infections, and alkalosis [11,12]. An additional application is the visual monitorization of urea in blood or urine to detect kidney failure. This failure provokes considerable increments in the release levels of urea and other metabolic bodily wastes. Combining a urease enzyme with a halochromic substance enables the analysis of the urease reaction through the pH increment of the solution as a result of the urea conversion into ammonia [13]. Several other sectors may benefit from the development of halochromic materials such as geotextiles or protective clothing that measure in real time pH alteration and detect soil or air contaminants. In architecture and civil engineering, these applications may be used to pinpoint material corrosion in steel structures [4,14]. These compounds may also be used to monitor the release of volatile amines resulting from bacterial growth in food. Being indicative of the quality/freshness, they can be incorporated in intelligent packaging that encloses meat, fish, and pasteurized products [15,16]. In addition, halochromic applications may be used to measure volatile acids and control the organoleptic characteristics of wine and other fermented food products [17]. It can also be applied in protective textiles to indicate acidic or alkaline vapors [18].

Halochromic textiles are mainly based on pH-sensitive dye molecules such as phthalides, triarylmethanes, and fluoranes. Few azo, styryl, and indophenol compounds provide a good color shift [19,20]. The halochromic molecules present weak acid or basic character, conjugated bonds, and are able to suffer changes upon protonation or deprotonation [21,22,23]. Traditional methods of textile coloration have been used to apply halochromic dyes such as exhaustion. It is a fairly economical method, where the dye molecules in the aqueous bath are adsorbed first onto the fiber surface and then diffused into the fiber. Depending on the fiber type, the pH of the dyeing bath and the affinity of the dye to the fiber are important parameters to avoid dye leaching [14,24].

Silk is a protein-based biopolymer with several benefits for biomedical applications, namely flexibility, permeability to oxygen and water, comfort, biocompatibility, controllable biodegradability, hemostatic and non-immunogenic properties, ability to function as a barrier to bacterial colonization, durability, dyeability, lightweight and outstanding mechanical properties [25,26]. These characteristics make silk a suitable material per se or in combination with other materials for drug-delivery systems, composite wound dressings, tissue engineering, regenerative medicine, water ultrafiltration systems, and biosensors [27,28,29,30].

In this work, for the first time, silk fabric was dyed by exhaustion with one of the recently developed halochromic dyes (AzoIz.Pip) [31]. The halochromic properties of the dye in solution, and in the silk fabrics, were assessed by recording the ultraviolet-visible (UV-vis) spectra, the color strength (K/S), color difference (∆E), CIE L*a*b* coordinates, and ultraviolet protection factor (UPF). The dyeing process was tested using different pH, temperatures, and dye concentrations. The halochromic performance was confirmed using the universal Britton–Robinson buffer from pH 3 to 12 and artificial wound fluids (acid and alkaline perspiration, and wound exudate). The stability of the dyes in the solutions and adsorbed in the fabric, reversibility, and fastness properties were also studied.

## 2. Materials and Methods

### 2.1. Materials

Commercial pre-washed silk fabric (weight per unit area of 57 g·m^−2^, warp/weft density of 50/140) was used. The fabric was washed at 40 °C for 60 min, rinsed with distilled water, and dried at 40 °C.

All the reagents used were of analytical grade, or of the highest grade available.

Diaminomaleonitrile (DAMN), 1,4-dioxane, triethyl orthoformate (TEOF), dimethylamine, phenylhydrazine, Macherey-Nagel™ aluminum sheets UV254 and 1,8-diazabicyclo [5.4.0]undec-7-ene (DBU) (Acros Organics, Thermo Fisher Scientific, Alfagene, Carcavelos, Portugal). Acetonitrile, silica gel, diatomaceous earth, anilinium chloride, 3-morpholinopropane-1-sulfonic acid (MOPS), phosphoric acid, boric acid, potassium chloride, creatinine, _D_-glucose, yeast extract, and haemin (Sigma Aldrich, Hamburg, Germany). Acetic acid (Chemlab, Zedelgem, Belgium); piperidine and urea (Riedel-de Haen, Charlotte, NC, USA); diethyl ether, *n*-hexane, silica gel 60, sodium chloride, sodium hydroxide, and peptone (Panreac, Barcelona, Spain). Deuterated DMSO (TCI, Zwijndrecht, Belgium). Absolute ethanol (VWR chemicals, Carnaxide, Portugal). ECE non-phosphate reference detergent (SDC Enterprises Ltd., Thongsbridge, UK).

Reagents used in cell culture, including Dulbecco’s modified Eagle’s medium (DMEM) with 4.5 g·L^−1^ glucose and GlutaMAX™, heat-inactivated fetal bovine serum (FBS), 0.25% trypsin/1 mM ethylenediaminetetraacetic acid (EDTA), antibiotic (10,000 U·mL^−1^ penicillin, 10,000 µg·mL^−1^ streptomycin), and Hanks’ balanced salt solution (HBSS) without calcium and magnesium [HBSS (−/−)] (Gibco^TM^, Thermo Fisher Scientific, Alfagene, Portugal). Neutral red (NR) solution, (4,5-dimethylthiazol-2-yl)-2,5-diphenyl tetrazolium bromide (MTT) and dimethyl sulfoxide (DMSO) (Sigma-Aldrich, Taufkirchen, Germany). Triton™ X-100 detergent solution (Thermo Fisher Scientific, Waltham, MA, USA).

### 2.2. Synthesis and Characterization of the AzoIz.Pip Dye

The (*E*)-(5-imino-1-methyl-2-(piperidin-1-yl)-1,5-dihydro-4*H*-imidazol-4-ylidene)((*E*)-phenyldiazenyl)methanamine (AzoIz.Pip) was prepared according to a recently reported method [31]. First, (*Z*)-5-amino-1-methyl-*N*′-phenyl-1*H*-imidazole-4-carbohydrazonamide (Amz) was prepared in a four-step synthetic pathway using the commercial reagents DAMN, TEOF, methylamine, and phenylhydrazine [32,33,34]. Then, (*E*)-5-amino-4-(imino(phenyldiazenyl)methyl)-1-methyl-1*H*-imidazol-3-ium (AzoIz) was obtained from the oxidation of Amz, and subsequent reaction of the AzoIz with piperidine gave the AzoIz.Pip. All compounds were characterized by ^1^H NMR to confirm their purity before further tests. The halochromic properties of AzoIz.Pip were studied using the Britton–Robinson buffer solutions from pH 3 to 12 (Table 1). Moreover, the behavior of AzoIz.Pip was also studied using artificial body fluids such as acid (pH 5.5) and alkaline (pH 8.0) sweat prepared according to ISO 105 E04:2013 and wound exudate (Table 2). The pKa of AzoIz.Pip in aqueous medium was determined by UV-vis spectrophotometry [35], using an initial solution of the molecule (64 μg·mL^−1^) to which 2.0 molar equiv. of HNO_3_ was added and diluted with the buffer solutions (1:1). Molar extinction coefficient was determined by the Beer–Lambert law equation (1), where “A” corresponds to the absorbance, “ε” corresponds to the molar extinction coefficient (L·mol^−1^·cm^−1^), “l” is the cell width (cm), and “c” is the concentration of AzoIz.Pip in the solution (mol·L^−1^).
(1)A= ε·l·c

### 2.3. Functionalization of Silk with AzoIz.Pip Dye under Different Conditions

All dyeing experiments were performed in an Ahiba Turbocolor dyeing machine (Datacolor, Lawrenceville, NJ, USA). The dyeing process was performed using solutions with different pH values (6 and 8) and temperatures (40 or 70 °C) for 60 min and at 40 rpm in a 1:20 ratio bath. Different dye concentrations were studied ranging from 4 to 128 μg·mL^-1^. The samples were rinsed with distilled water and dried at 40 °C.

### 2.4. pH and Color Measurements

pH measurements were performed with HANNA instruments HI5221-02 (HANNA instruments, Póvoa de Varzim, Portugal) with a pH HI1131B electrode and an HI7662-T temperature probe. UV-vis spectra were collected in a Shimadzu UV-1800 (Shimadzu Europa GmbH, Duisburg, Germany) using 1 cm wide polymethyl methacrylate cuvettes. For the reflectance and transmittance measurements on the textile samples, a Shimadzu UV-2600 spectrophotometer (Shimadzu Europa GmbH, Duisburg, Germany) was used. The spectra were recorded from 280 to 800 nm with an interval of 1 nm using a D65 illuminant and a standard observer of 10 degrees. The color of the silk samples was evaluated in the Commission Internationale de l’Elcairage (CIE) L* (lightness), a* (transition from green (−a*) to red (+a*)), and b* (transition from blue (−b*) to yellow (+b*)) space using RGB values obtained using the program UV-2401PC Color Analysis (color-shortcut). The transmittance data were used to calculate the UPF, applying the AATCC test method 183-2020. ∆E was determined according to equation (2), where “∆L*” is the lightness difference, “∆a*” is the redness–greenness difference, and “∆b*” is the yellowness–blueness difference. ΔE > 1 reveals a visually detectable color difference [38]. K/S of the dyed silk samples was assessed using the reflectance method and Kubelka–Munk equation (3), and by measuring twice the reflectance at different positions of the fabric, where “K” is the absorption coefficient, “S” scattering coefficient, and “R” the reflectance at complete opacity [39].
(2)$$E=\sqrt{{\left(\Delta L^{*}\right)}^{2}+{\left(\Delta a^{*}\right)}^{2}+{\left(\Delta b^{*}\right)}^{2}}$$
(3)$$K/S={\left(1-R\right)}^{2}/2R$$

### 2.5. Evaluation of the Halochromic Properties of the Dyed Silk Samples, Stability of the Dyes in Silk (Leaching and Color Change in Different Time Periods), and Reversibility of Color Change

The sensitivity of the dyed silk to pH variations was evaluated by dipping samples (2 cm × 3 cm) in Britton–Robinson buffers from pH 4.0 to 10.0 and in body artificial fluids (wound exudate at pH 7, sweat at pH 6 and 8) for 1 h, drying it at 40 °C, and performing the color assessment. Moreover, samples dyed with the different concentrations of AzoIz.Pip under acidic conditions at 40 °C were immersed in a Britton–Robinson solution at pH 10, and the color was characterized to detect color differences using diverse concentrations.

The stability of the dyes in silk samples was also tested by dipping the silk samples in Britton–Robinson buffer solutions (pH 5, 7, 8, 9, and 10) and removing them at different times (after 1, 6, and 24 h of dipping). The samples were dried and the color assessment was performed. Here, the remaining buffer solutions were also analyzed by UV-vis to evaluate the leaching of the dye during the immersion time.

The reversibility of color change was assessed four times by dipping four initial silk samples (2 cm × 3 cm), prepared under acidic conditions at 40 °C, in a Britton–Robinson solution at pH 10 at the same time. The samples were kept in the solution for 10 min. Then, one of the samples was dried to observe the first color change and the other three samples were immersed in a Britton–Robinson solution at pH 6 simultaneously. After 10 min, one of the samples was dried to detect the second color variation, and the other two samples were dipped into another solution of Britton–Robinson at pH 10. After 10 min of contact, one of the samples was removed and dried to characterize the third color change. The last sample was dipped 10 min more in a Britton–Robinson solution at pH 6, removed, and dried to observe the fourth color change. The CIE Lab color coordinates, ∆E, K/S, and UPF were tabled to detect all color variations, and the remaining buffer solutions were analyzed by UV-vis to evaluate the leaching of the dye during this experiment.

### 2.6. Fastness Studies

#### 2.6.1. Domestic and Commercial Washing

A color fastness to domestic and commercial washing assay was performed following the standard ISO 105-C06:2010 in a Washtec—P equipment (Roaches, West Yorkshire, UK), and a multifiber adjacent fabric (DW) was used according to ISO 105-F10:1989. Briefly, a 4 cm × 10 cm sample was sewed to the DW. The fabrics were immersed in a 150 mL washing bath containing 4 g·L^−1^ of ECE detergent and 10 stainless steel spheres. The assay was conducted for 30 min, at 40 °C (assay number—A1S). The color fastness was evaluated with UV-vis analysis of the tested sample (ΔE determination) and with a grey scale for assessing the staining of the DW.

#### 2.6.2. Rubbing (Dry and Wet)

Color fastness to rubbing was evaluated according to the standard ISO 105-X12:2016. The samples measuring 5 cm × 14 cm were placed in a crockmeter (Roaches, West Yorkshire, UK) sample holder, with a wet (distilled water) or dry 5 cm × 5 cm cotton rubbing cloth (according to ISO 105-F09:2009) placed in the test head. The cotton cloth was rubbed 10 cycles, applying a pressure of 9 N, with a velocity of one cycle per second. The color fastness was evaluated with ΔE determination and a grey scale for assessing the staining of the cotton cloth fabric.

#### 2.6.3. Perspiration and Wound Exudate

Color fastness to perspiration was performed according to ISO 105-E04:2013; two solutions were analyzed—acid perspiration (pH 6) and alkali perspiration (pH 8). Briefly, samples measuring 4 cm × 10 cm were sewed to the DW and were immersed into the solutions in a 1:50 ratio bath at room temperature for 30 min. Then, the specimens were passed through two glass rods, to remove the excess solution. The samples were laid between two glass plates and placed in the perspirometer (Roaches, West Yorkshire, UK) with a pressure of 49 N. The equipment was placed into the oven at 37 °C for 4 h in the horizontal position. The samples were dried at room temperature. The color fastness was evaluated by ΔE determination and with a grey scale for assessing the staining of the DW. The same procedure was implemented using the initial samples immersed in the artificial wound exudate solution prepared according to Table 2.

### 2.7. X-ray Photoelectron Spectroscopy (XPS)

Detailed surface atomic composition and bonding environment research was conducted employing an XPS PHI-TFA spectrometer (Physical Electronics Inc., Chanhassen, MN, USA) equipped with an Al- monochromatic (7 mm) X-ray source operating at pass energy equal to 1486.6 eV, with active surface charge neutralization. Data acquisition was performed with a vacuum better than 1 × 10^−8^ Pa. The charge of the spectra have been corrected to give the adventitious C1s spectral component (C–C, C–H) binding energy of 284.5 eV. Spectra were analyzed for elemental composition using Multipack software.

Deconvolution into sub-peaks was performed by OriginLab software, using the Gaussian fitting function and Shirley-type background subtraction. No tailing function was considered in the peak fitting procedure.

### 2.8. FTIR-ATR

The spectra were recorded with a PerkinElmer (USA) spectrometer in ATR mode. The study interval ranged from 3900 cm^−1^ to 600 cm^−1^ with a spatial resolution of 2 cm^−1^ and accumulated 10 scans from 5 different spots, which were averaged to a final spectrum.

### 2.9. Cytotoxicity of AzoIz.Pip-Dyed Silk Extracts

The cytotoxicity of textile extracts towards HaCaT cells was evaluated, 24 h after exposure, by the NR uptake and MTT reduction assays [40]. HaCaT cells, an immortalized human keratinocyte cell line, were obtained from the American Type Culture Collection (ATCC; Manassas, VA, USA) and were routinely cultured in 75 cm^2^ flasks using DMEM with 4.5 g·L^−1^ glucose and GlutaMAX™, supplemented with 10% heat-inactivated fetal bovine serum (FBS), 100 U·mL^−1^ penicillin, and 100 μg·mL^−1^ streptomycin. Cells were maintained in a 5% CO_2_–95% air atmosphere, at 37 °C, and the medium was changed every 2–3 days. Cultures were passaged weekly by trypsinization (0.25% trypsin/1 mM EDTA).

For the cytotoxicity evaluation of AzoIz.Pip-dyed silk extracts, the cells were seeded in 96 well-plates at a density of 60,000 cells.cm^−2^ and, 24 h after seeding, the cell culture medium was removed, and the cells were exposed to different concentrations of the textile extracts (0–100%) for 24 h. For each assay, extracts of each sample of textiles were freshly prepared, according to ISO 10993-5 (Biological evaluation of medical devices—Part 5: Tests for in vitro cytotoxicity). Briefly, the extraction was performed in a complete cell culture medium, at a proportion of 0.1 g·mL^−1^, in a sterile, chemically inert, and closed container for 24 ± 2 h at 37 ± 1 °C under agitation. The extract was then directly used (100% concentration) or diluted in fresh cell culture medium at different concentrations (50, 25, 12.5, 6.25, and 3.125%). Extraction cell culture medium (without the test material) was also submitted to the same extraction conditions and used as a control. Triton™ X-100 (1%) was used as a positive control. The cells used in all experiments were taken between the 57th and 63rd passages.

#### 2.9.1. Neutral Red Uptake Assay

The cytotoxicity of the samples was evaluated by the NR uptake assay, in which the estimation of viable cell numbers was assessed based on their ability to incorporate and bind the supravital dye NR in the lysosomes. At the selected time point (24 h), the cell culture medium was removed and replaced by fresh cell culture medium containing 50 μg·mL^−1^ NR. The cells were then incubated at 37 °C in a humidified 5% CO_2_–95% air atmosphere for 40 min. After the incubation period, the cell culture medium was removed, followed by the extraction of the dye absorbed only by viable cells with absolute ethyl alcohol/distilled water (1:1) with 5% acetic acid. The absorbance was then measured at 540 nm in a multiwell plate reader (PowerWaveX BioTek Instruments, Winooski, VT, USA). The percentage of NR uptake relative to control cells (0 µM) was used as the cytotoxicity measure. Results were obtained from four independent experiments, performed in triplicate.

#### 2.9.2. MTT Reduction Assay

The cytotoxicity of the samples was evaluated by the MTT reduction assay, in which the cellular metabolic capacity was assessed based on the ability of mitochondria living cells to cleave the tetrazolium ring of MTT forming a colorimetric formazan product. At the selected time point (24 h), the cell culture medium was removed and replaced by fresh cell culture medium containing 0.5 mg·mL^−1^ MTT. The cells were then incubated at 37 °C in a humidified 5% CO_2_–95% air atmosphere for 40 min. After the incubation period, the cell culture medium was removed, and the formed formazan crystals dissolved in 100% DMSO. The absorbance of the formed formazan was measured at 550 nm in a multi-well plate reader (Power Wave-X, BioTek Instruments, Winooski, VT, USA). The percentage of MTT reduction relative to control cells (0 µM) was used as the cytotoxicity measure. Results were obtained from four independent experiments, performed in triplicate.

#### 2.9.3. Statistical Analysis

All the statistical assessments were performed using the GraphPad Prism 8 for Windows (GraphPad Software, San Diego, CA, USA). Two-way ANOVA was used to perform the statistical comparisons, followed by Sidak’s multiple comparisons test.

## 3. Results and Discussion

### 3.1. AzoIz.Pip Synthesis and Corresponding Halochromism in Aqueous Solution from pH 3 to 12

Recently, a novel method to obtain 2-aminoimidazole azo dyes with distinct substituent patterns was developed by our research group, showing interesting functional properties [31,33,41]. These emergent molecules were obtained by oxidation of the Amz and subsequent reaction with secondary amines such as piperidine to obtain a novel azo dye (AzoIz.Pip) (Figure 1). The AzoIz.Pip showed vibrant colors and attractive halochromic properties, showing a hypsochromic shift from blue (561 nm) to magenta (541 nm) when the base was added to mixtures of water/ethanol (80:20) [31,33]. Thus, AzoIz.Pip dye is negatively halochromic. It is important to note that negative halochromism is not a common phenomenon in azo dyes, and it is detected only in cases where exceptional acceptors or donor moieties are present [42].

The universal aqueous Britton–Robinson buffer was used to evaluate the color variation in AzoIz.Pip dye in pH ranging from 3 to 12. The UV-vis spectrum of the dye obtained in this aqueous medium was compared with the previously reported results in a water/ethanol (80:20) mixture. It was found that the halochromic properties change with the composition of the medium. In this buffer, maximum absorption peaks were detected at 557 nm when the pH changed from 3 to 8, and the shift to 544 nm occurred at pH 9. A higher shift was even observed for pH equal to or higher than 10, as the absorption band changed to 536 nm (Figure 2a). The pKa values were achieved by UV-vis spectrophotometry by determination of the wavelengths of the maximum absorption peak in the first shift (pH 3 to 8), as well as in the second shift (from pH 9 to 12). Thus, the two extreme pH values of each shift were considered, and a plot of the variation of the absorbance according to the pH is displayed in Figure 2b. Two pKa values were found, 8.40 for pKa1 and 9.52 for pKa2, by determining the pH of the corresponding intersection points. ε were determined, and it was observed that the ε values increased with pH, which may be associated with a higher concentration of the deprotonated form of the molecule in the solution (Table 3). The ε increased from 5.79 × 10^3^ L·mol^−1^·cm^−1^, in pH values below the pKa1, to 6.43 × 10^3^ L·mol^−1^·cm^−1^ in pH values between pKa1 and pKa2, and even higher values were obtained at pH above pKa2 (ε = 8.03 − 8.53 × 10^3^ L·mol^−1^·cm^−1^). Moreover, the molecule’s stability in solutions of different pH values was assessed by collecting the UV-vis spectra at 7 h (Figure 2c) and 24 h (Figure 2d) after solution preparation. A stable absorption band was observed at pH values below the pKa1. However, the maximum absorption bands considerably decreased after 7 h at pH above pKa2, and almost disappeared after 24 h in pH ranging from 10 to 12. Thus, the dye decolorization is slower at pH 9 (between pKa1 and pKa2), but the color is drastically lost above this value. This can be explained by degradation reactions that occur when the compound is maintained in aqueous solutions under strongly alkaline conditions.

### 3.2. Dyeing Optimization of Silk by Exhaustion under Different Bath, Temperatures, and Dye Concentration

A traditional exhaustion dyeing process was performed, and the effects of pH, temperature, and dye concentration were analyzed. The affinity of a dye to a fabric is strongly dependent on the pH, the charge, and available functional groups on the silk fabric. Firstly, the dye concentration of 64.0 μg·mL^-1^ was used in the dyeing process at pH 6 and 8 in two different temperatures, 40 and 70 °C. pH 6 and 8 were tested since they are above the isoelectric point of silk (about 4.4), and below the pKa1 of the molecule [43]. At these pH values, the silk fabric presents a high negative charge and the AzoIz.Pip dye presents a positive charge. Therefore, the dye may be easily adsorbed into the fiber surface by electrostatic interactions. In addition, temperature is also an important parameter in dyeing processes, as higher temperatures promote a better diffusion of the dyes in the fabrics [44]. In parallel with the silk dyeing process, solutions containing only the dye were placed exactly under the same conditions to be used as controls for the dye stability under the different conditions and to predict the concentrations of dye loaded into the fibers (Figure 3a). Using these control solutions, it was possible to detect a decrease in the maximum absorption band both at the temperature of 70 °C and pH 8, which was attributed to the superior hydrolysis of the dye under higher temperatures and alkaline conditions. The UV-vis spectra were also acquired from the solutions obtained after the exhaustion process with silk fabric, and it was observed that complete dye adsorption occurred, independently of the pH and temperature (Figure 3a). Regarding the silk-dyed samples, the K/S spectra showed the maximum value at 590 nm in all conditions tested. However, the intensity of this band was considerably higher when conditions of pH 6 and 40 °C were used, which is in accordance with UV-vis spectra (Figure 3b).

In the color coordinates, the L* value increased when pH and temperature increased, indicating a lower concentration of the dye in the fabrics, or superior dye degradation in these conditions. The K/S sum and ∆E showed higher values for samples dyed at pH 6 and temperature of 40 °C when compared with the control (Table 4), confirming the superior stability of the dye in these conditions. Therefore, silk samples dyed at pH 6 and 40 °C were used in further analysis.

Several factors are related to the UV-blocking performance of the fabrics, including the fiber type, applied pre-treatments, fabric construction (thickness and density), and functionalization with dyes containing aromatic moieties [45]. The functionalization of silk with AzoIz.Pip was shown to slightly increase the UPF value, but a weak performance was achieved due to the fabric structure and thickness (Table 4) as the UPF value increased from 5 to 7–9.

Another experiment was performed to detect any color variations with the use of different concentrations of dye. The optimized exhaustion process was performed with concentrations ranging from 4 to 128 μg·mL^−1^. As expected, higher concentrations of AzoIz.Pip produced a stronger coloration (Figure 4 and Table 5). The UV-vis spectra showed that the same peak at 590 nm dominated in all samples, thus giving a bluish color to the silk. A complete exhaustion percentage was obtained even when higher concentrations were used (100% of exhaustion in all concentrations tested).

### 3.3. Evaluation of the Halochromic Properties of Dyed Silk

The dyed silk fabrics were dipped into different buffered Britton–Robinson solutions (from pH 4 to 10) and in artificial body fluids (sweat at pH 6 and 8, and wound exudate at pH 7) to determine the halochromic properties of the samples (Figure 5 and Table 6). Similar to dyes in solution, a clear color change was visually perceived in the textile samples that were immersed in solutions with pH 9 and 10. Despite the samples being immersed for 1 h in the solutions, the response time was very fast, being detected with the naked eye for pH above 8. Within a few seconds, the silk samples acquire their final magenta color. In general, the *b value decreased and the ∆E and a* values increased by increasing the pH, proving the color variation. Comparing the blue sample dipped in the Britton–Robinson solution at pH 4 with the one at pH 9, the color coordinates indicate the color change as the b* (redness/greenness) value decreased from −16.73 to −20.02. When compared with the initial dyed sample, the highest value in the color difference was obtained at pH 10 (9.16). Despite some differences observed in color coordinates for the assays performed with the artificial body fluid buffers, the color of these samples was compared with the colors obtained at the same pH values using the Britton–Robinson solution, and a similar blue coloration is perceived with the naked eye. The stability of the color under distinctive environments was confirmed since the composition of these fluids is much more complex and unpredictable. In addition, a slight variation was observed in the K/S value. Similarly to the behavior of the halochromism in solution, the silk samples in alkaline conditions also showed a stronger color, which is reflected in its K/S, which increased from 0.54 to 0.58. The K/S spectra revealed a hypsochromic shift when pH increases, as the maximum value shifted from 590 (until pH 8) to 570 (pH 9) or 560 nm (pH 10) (Figure 5). The effect of concentration on the maximum wavelength value at pH 10 was achieved by dipping all the samples presented in Figure 4 (samples prepared at pH 6 in concentrations ranging from 4 to 128 μg·mL^−1^) in the buffer solution at pH 10 (Figure 6 and Table 7). The wavelength of the K/S maximum value was found at 560 nm in all samples. By increasing the concentration, the K/S sum and ∆E value were raised from 0.33 to 0.77 and from 4.48 to 44.61, respectively, proving that the color tone is the same and independent of the concentration.

### 3.4. Stability of the Halochromic Dyes in Silk Fabric and Leaching Evaluation

The stability of the dyes adsorbed in the silk samples and leaching was assessed by submerging the initial dyed sample at pH 6 in Britton-Robbinson buffer solutions at pH 5, 7, 8, 9, and 10. Subsequently, silk samples were removed from the solution after 1, 6, and 24 h. The stability and leaching were measured by recording the UV-vis spectra of the buffer solutions and also by recording the color of the samples (Figure 7 and Table 8).

From the UV-vis spectra of the buffer solutions, any leaching was detected. However, in the samples immerged at pH 8, 9, and 10, a slight decrease in the K/S maximum value was noticed, mainly after 24 h, suggesting some leaching or dye degradation at these pH values. Moreover, the *L value also showed a slight increase in samples dipped in buffers with pH equal to or superior to 7. Despite this, after 24 h under strongly alkaline conditions, the coloration of the fabric was kept obvious, in contrast to the results obtained with the molecules in solution. Thus, the silk fabric was shown to stabilize the AzoIz.Pip molecule, avoiding its complete degradation after 24 h.

### 3.5. Study of the Reversibility of the Color Change in the Silk Fabrics

The reversible effect of halochromic dyes in silk samples was tested for four cycles (Table 9). The samples were dipped in solutions at pH 6 or 10 in intervals of 10 min. The ∆E value as well as the CIE Lab color coordinates confirmed the reversibility. The K/S sum value does not display significant differences, confirming that no release occurs during color changes. Moreover, the ∆E values are similar between cycles. The first and third cycles revealed an ∆E of 8.44 and 8.28, and the second and fourth cycles an ∆E of 1.87 and 2.12, compared with the initial dyed samples. The similar values indicated that the halochromic silk fabrics had high stability within the halochromic pH range.

### 3.6. Colorfastness Tests

Fastness tests were performed on the samples dyed under optimal conditions in terms of color yield. Fastness to washing, rubbing (wet and dry), and perspiration at acidic and alkaline pH were assessed, and also the standard of perspiration was adapted using the artificial wound exudate (Table 10). The silk samples showed medium/good fastness results regarding the staining of the multifiber adjacent fabric. An insignificant dye transfer from the silk samples was observed in almost all fastness tests (Table 10). Some dye was transferred only for acetate and polyamide 6,6 fibers, revealing staining of 3–4 using the grey scale. ∆E values showed poor resistance of the color to washing, as a ∆E of 14.58 was obtained when comparing the final sample with the silk dyed at pH 6 and 40 °C, indicating that these halochromic silk samples can only be applied to disposable or non-washable products. The additional ∆E values were not so substantial, only for alkaline perspiration, which decreases by 1.30. The remaining ∆E data showed values less than one. The ∆E variations were not attributed to the leaching, but to the stability of the dye in the fiber. As observed in the study of solutions by UV-vis, the dye loses color in solution at a pH equal to or above nine, and the color lost is soon detected after 6 h. So, in fastness tests for alkaline perspiration and washing, the discoloration can be attributed to the degradation of the dye promoted either by the pH alkaline or by the action of the ECE detergent (only used in the washing fastness test). The leaching option is not admitted due to the absence of bands in the UV-vis, either in the stability test (Section 3.4) or the reversibility test (Section 3.5). Still, silk has been shown to stabilize the basic form of the dye since the color variation is completely noticeable after 24 h, as demonstrated in Section 3.4.

### 3.7. XPS Analysis

Chemical analysis by photoelectron spectroscopy was undertaken to investigate the interaction between silk fibers and AzoIz.Pip dyes under slightly acidic ‘pH:6’ (below pKa1 = 8.40) and alkaline ‘pH:10’ (above pKa2 = 9.52) environments. First, the survey spectra were collected (Figure 8a–c), from which the contribution of the C1s, N1s, and O1s peaks was figured out. Naturally, a silk fiber consists mainly of repeating sequences of glycine (about 40%), alanine (30%), serine (10%), and tyrosine (5%), characterized by a dominant proportion of carbon and a similar amount of nitrogen and oxygen atoms (hydrogen is not directly detectable by XPS) [46]. The calculated elemental shares for a control sample are C1s = 71.8 at. %, O1s = 15.5 at. %, and N1s = 12.7 at. % (Figure 8a). The dyed sample did not change significantly after immersion in a slightly acidic environment (no degradation is expected) (Figure 8b). It shows a slight decrease in carbon content and a tiny increase in oxygen share (C1s = 70.3 at. %, O1s = 16.8 at. %, and N1s = 12.9 at. %). A visual change occurs in a sample that has interacted with a strong alkali solution (NaOH), with the carbon peak in the spectrum showing a lower intensity and the O1s peak predominating (C1s = 66.3 at. %, O1s = 19.4 at. %, and N1s = 13.0 at. %). A trace of sodium (Na1s = 1.3 at. %) was also observed as a result of interaction with the base. A change in the surface chemistry might be related to alterations happening in the dye or/and in the molecular structure of the silk. To answer this question, Gaussian deconvolution was performed for the core level C1s peaks (Figure 8a’–c’ and Table 11) [47].

The resulting deconvolution revealed an interesting behavior of the carbon surface components, but let us define them first. A set of four components was selected to fit the normalized C1s profile, with C1 at 284.5 ± 0.1 eV standing for C-C/CH, C2 at 285.5 ± 0.1 eV mainly representing the C-N bond, C3 at 286.5 ± 0.1 eV representing C-O bond, and last but not least, the C4 peak at 288.0 ± 0.1 eV attributed to C=O (Figure 8a’–c’). After examining their fitted area, it can be suggested that the C3 component is most altered with respect to a control silk sample. An increased contribution of the C3 component could explain an increase in oxygen content after interaction with acidic and alkaline aqueous solutions. A relative change in the other three peaks is much smaller, making it more problematic to explain their behavior. In order to find a reasonable explanation, the N1s and O1s core levels were analyzed by examining residual spectra (Figure 9).

From the data shown in Figure 9b and 9b’, it can be said that an alkaline environment exerts a stronger chemical effect on the NH-containing bonds in N1s core level and strongly alters the doubly bonded oxygen in the carboxyl group (O**=C-OH). Note that before subtraction, the intensity of the peaks (high-resolution regime) was normalized, so that the spectral difference between the control and ‘pH:10’ is even larger (see O1s peaks in Figure 9a’,b’). In contrast, a slightly acidic environment affects the oxygen and nitrogen core regions to a lesser extent. The surface chemistry alterations (dye layer + silk surface) were associated with structural changes (deeper into silk), which were investigated by FTIR (Figure 10). In the high-frequency stretching vibrations region, the spectra are quite similar, and in the C-H stretching vibrations region (symmetric and asymmetric modes), there is a slight reduction in intensity for a sample with a pH of 10, which could be related to a reduction in carbon content. A tiny gain can be spotted in OH-stretching vibrations (positive feature at about 3600 cm^−1^ in Figure 10c), which agrees well with somewhat increased oxygen content in the XPS spectra at both pH 6 and pH 10 conditions.

In the fingerprint area (Figure 10b), there is a negligible difference between the control and pH 6 samples, indicating the molecular integrity of the latter and the absence of noticeable degradation. However, the situation is different at pH 10, where the bands of amide I (mainly associated with C=O stretching vibrations) and amide II (secondary N-H bending) are most altered, with the first band increasing in alkaline media (and also shifting to a lower wavenumber from the original 1630 cm^−1^ to 1620 cm^−1^), and the second losing its intensity (with upwards shifting from 1505 cm^−1^ to 1512 cm^−1^) [48,49,50]. The bands of the amide III, which typically represent C-N stretching vibrations and in-plane N-H bends, are not strongly affected. In addition, minor features attributable to C-O-C (decrease) and C-O stretching (increase) are also altered. Their trend follows the XPS inference: less carbon and more oxygen after the sample is immersed in alkaline pH:10 media. Moreover, prolonged contact with alkali leads to a deformation of the ring mode vibrations, which are located between 650 and 700 cm^−1^. Still, according to the data explained previously, it can be said that the silk is able to withstand more aggressive environments than the dye alone, and once the AzoIz.Pip molecule is absorbed into it, the stability of the dye is improved.

### 3.8. Cytotoxicity Evaluation of the Dyed Silk Extracts

The cytotoxicity of the AzoIz.Pip-dyed silk extracts was assessed, 24 h after exposure, by the MTT reduction and NR uptake assays (Figure 11). According to ISO 10993-5, if cell viability is reduced to <70% of the control cells, the medical device has a cytotoxic potential.

Significant effects on MTT reduction were detected for extract concentrations equal to or above 50%, with MTT reduction significantly decreasing to 51.4% and 9.6% when cells were exposed, for 24 h, to the 50% and 100% concentrations of extracts of silk dyed with AzoIz.Pip, respectively, and when compared to control cells.

Concerning the NR uptake assay, a significant reduction in the cell viability can also be observed when compared to the control cells, although the cytotoxic effect was only detected for the highest extract concentration (NR uptake significantly decreased to 62.0%, 24 h after exposure to 100% AzoIz.Pip- dyed silk extract).

These two assays are based on different physiological endpoints. MTT assesses the metabolic compromise, which normally occurs in the first instance, while the NR assesses cellular integrity. Therefore, under the present experimental conditions, the MTT reduction assay was demonstrated to be more sensitive in addressing the cytotoxic effects of AzoIz.Pip-dyed silk extracts. However, in both assays, a significant decrease in cell viability was observed, suggesting a cytotoxic potential for the extract obtained from silk dyed with AzoIz.Pip in the concentration of 64.0 μg·mL^−1^. The concentration of AzoIz.Pip in the textile should thus be decreased to non-cytotoxic values without compromising the halochromic properties.

## 4. Conclusions

In summary, this research described a simple and affordable method to produce unusual halochromic textiles taking advantage of the flexibility, biocompatibility, and lightness of silk, as well as the distinct halochromic properties of the AzoIz.Pip molecule, which changes color from blue to magenta as the pH increases. Thus, it was possible to develop, by the exhaustion method at low temperatures, an atypical pH-responsive material. The response was observed both in buffer solutions, universal Britton–Robinson buffer, and artificial body fluids, and in the silk-dyed samples. The conjugation of the silk fabric with the AzoIz.Pip promoted additional stabilization of the dye, especially at higher pH values. No leaching was detected by UV-vis spectrophotometry. The reversibility of the color change was assessed, demonstrating good results for at least four cycles. The fabric samples presented good results for perspiration, wound exudate, and rubbing fastness properties. The biocompatibility of the silk extracts was tested in HaCaT cells. The viability was reduced to <70% only when the dye concentration in the fabric was higher or equal to 64 μg·mL^−1^. The halochromic effects can be visually detected using lower concentrations of the dye in the fabric (from 8 to 32 μg·mL^−1^). Silk fabrics functionalized with the AzoIz.Pip can be useful in several smart materials applications including biomedical, sport, and protective clothing and packaging.

## Figures and Tables

**Figure 1 polymers-15-01730-f001:**
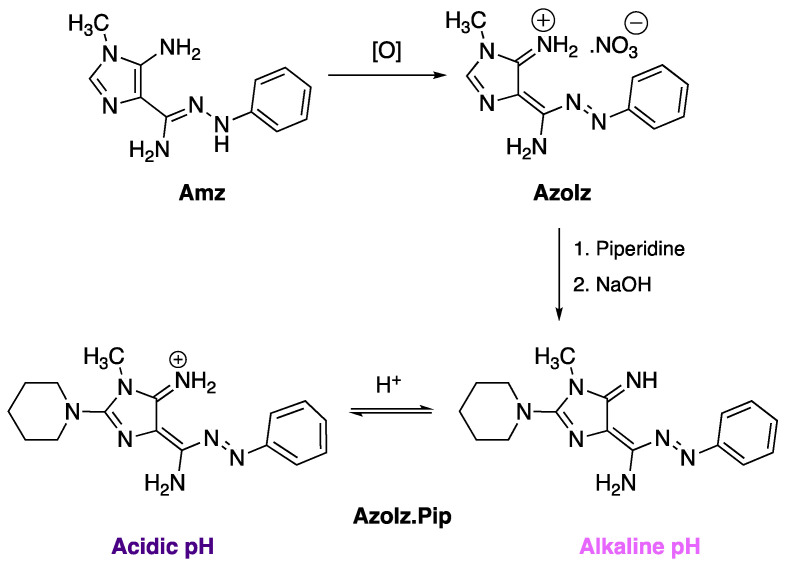
Synthetic route to obtain the AzoIz.Pip compound and corresponding protonation/deprotonation responsible for the color variation under different pH values.

**Figure 2 polymers-15-01730-f002:**
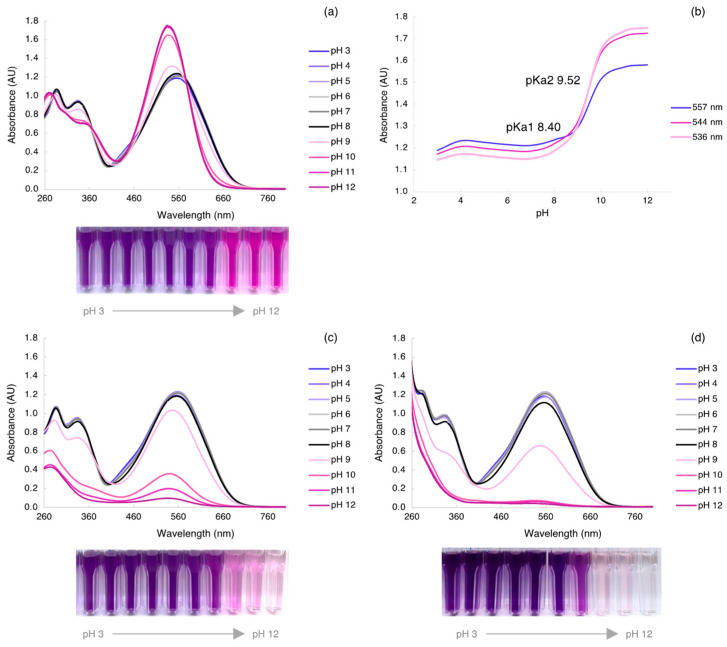
(**a**) UV-vis spectra of AzoIz.Pip from pH 3 to 12 immediately after solution preparation and (**b**) corresponding pKa calculation in the function of pH at 557 nm (λ_max_ at pH 3), 544 nm (λ_max_ at pH 9), and 536 nm (λ_max_ at pH 12); UV-vis spectra of the solutions (**c**) after 7 h and (**d**) 24 h of the preparation.

**Figure 3 polymers-15-01730-f003:**
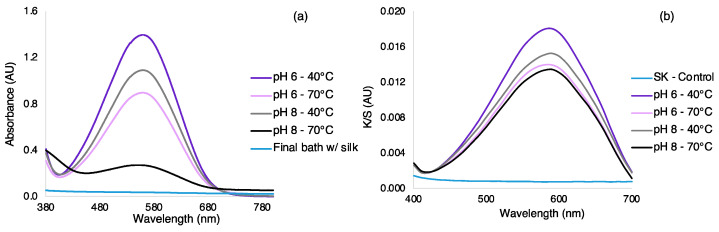
(**a**) UV-vis spectra of the control AzoIz.Pip solutions (1:2 diluted) (pH 6 and 8 and temperature at 40 and 70 °C) and UV-vis spectrum of the final bath of the exhaustion process with silk fabric, showing the complete adsorption of the dye; (**b**) K/S spectra of silk fabrics dyed with AzoIz.Pip dye using different exhaustion conditions.

**Figure 4 polymers-15-01730-f004:**
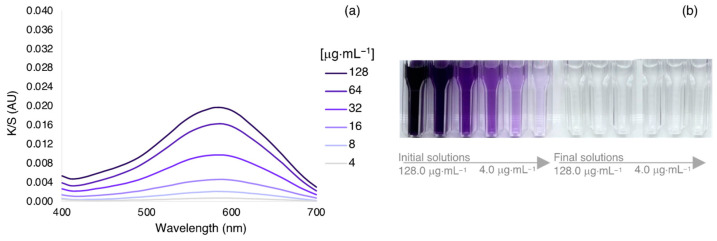
(**a**) K/S spectra of silk fabrics dyed with different concentrations of AzoIz.Pip dye under pH 6 and 40 °C, and (**b**) solutions of the different concentrations tested before and after the exhaustion process.

**Figure 5 polymers-15-01730-f005:**
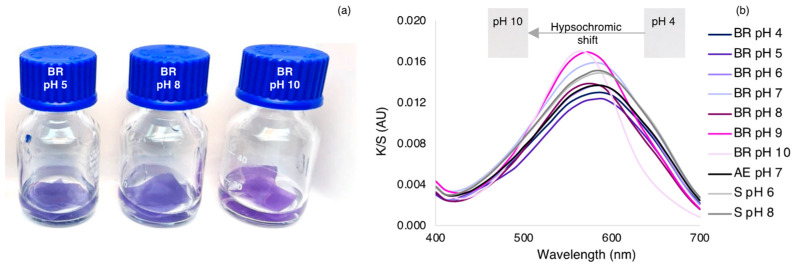
(**a**) Silk samples dyed at pH 6 dipped in different buffer solutions and corresponding color change at higher pH; (**b**) K/S spectra of silk fabrics dyed at pH 6 and dipped for 1 h in different buffer solutions (BR, Britton–Robbinson, AE–artificial wound exudate, S–artificial sweat).

**Figure 6 polymers-15-01730-f006:**
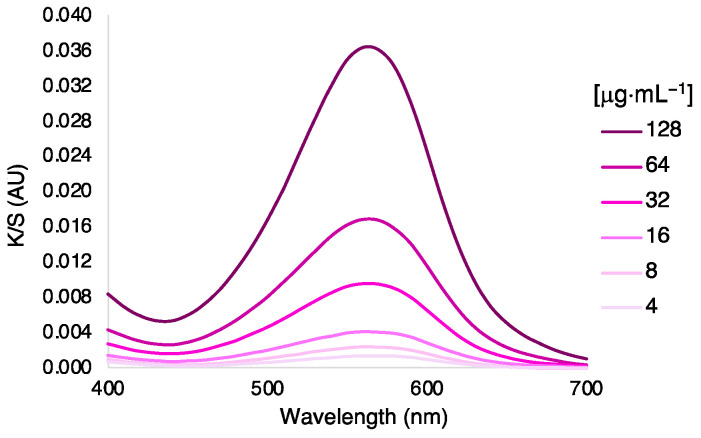
K/S spectra of silk fabrics dyed with AzoIz.Pip in different concentrations at pH 6 and immersed in a Britton–Robinson solution at pH 10.

**Figure 7 polymers-15-01730-f007:**
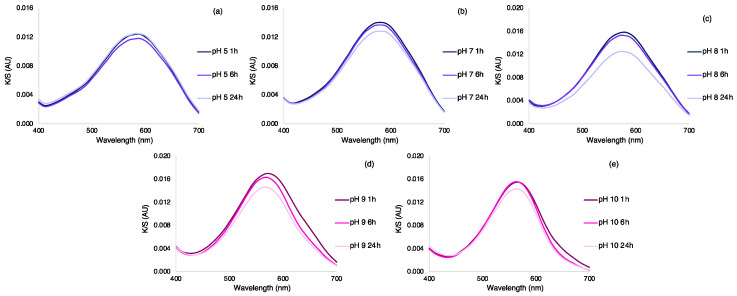
K/S of the silk samples after immersion in a Britton–Robinson solution at (**a**) pH 5, (**b**) pH 7, (**c**) pH 8, (**d**) pH 9, and (**e**) pH 10 for 1, 6, and 24 h (final samples of the stability test).

**Figure 8 polymers-15-01730-f008:**
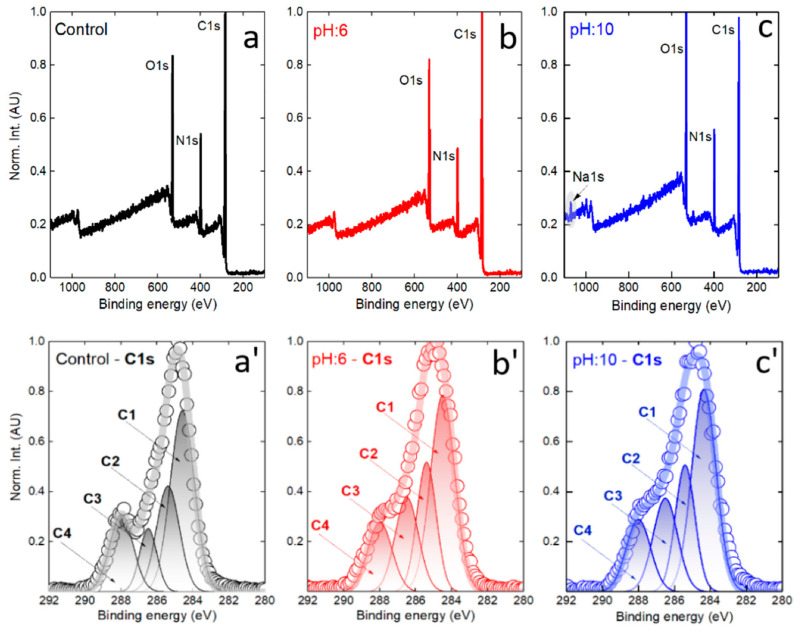
XPS survey spectra for control silk (**a**), dyed silk after interaction with acidic media ‘pH:6 (**b**) and alkaline environment (**c**). Gaussian fits of the corresponding C1s core level peak (**a’**–**c’**).

**Figure 9 polymers-15-01730-f009:**
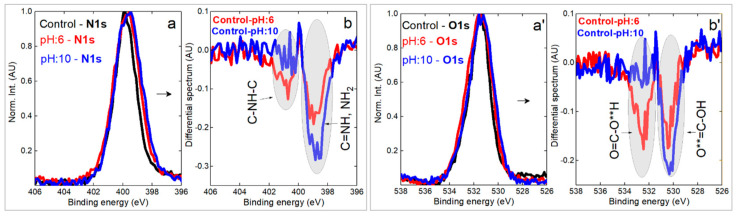
XPS core level N1s (**a**) and O1s (**a’**) spectra for control silk (black), dyed silk after interaction with acidic media ‘pH:6’ (red) and after influence of alkaline conditions ‘pH:10’ (blue) accompanied by corresponding residuum spectra (**b**,**b’**).

**Figure 10 polymers-15-01730-f010:**
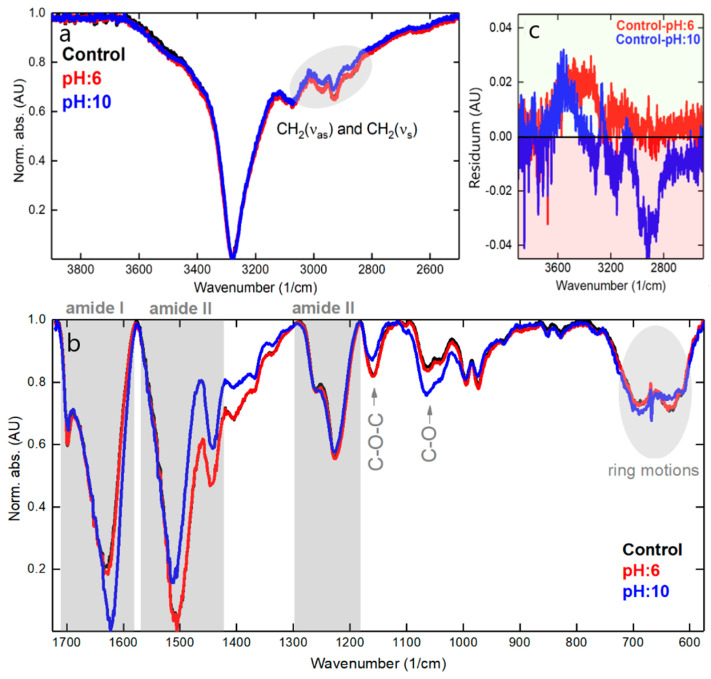
FTIR spectra (**a**,**b**) for the selected samples accompanied by residuum graph (**c**) for CH, NH, and OH stretching.

**Figure 11 polymers-15-01730-f011:**
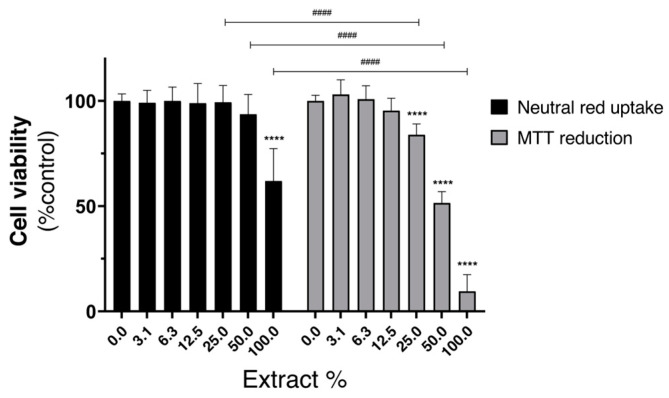
Cytotoxicity of AzoIz.Pip-dyed silk samples extracts (0–100%) evaluated in HaCat cells by the MTT reduction and NR uptake assay, 24 h after exposure. Results are expressed as Mean + SD from 4 independent experiences, performed in triplicate. Statistical comparisons were made using Two-way ANOVA followed by the Sidak’s multiple comparisons test (**** *p* < 0.0001 vs. 0%; #### *p* < 0.0001 for comparisons between cytotoxic assays, at each extract concentration).

**Table 1 polymers-15-01730-t001:** Britton–Robinson buffers composition, according to the method reported by Mongay et al. [36].

pH	[NaOH] (g·L^−1^)	[CH_3_COOH] (g·L^−1^)	[H_3_PO_4_] (g·L^−1^)	[H_3_BO_3_] (g·L^−1^)
3	1.04	2.09	3.409	2.15
4	1.60	1.92	3.136	1.98
5	2.07	1.78	2.904	1.83
6	2.39	1.69	2.751	1.74
7	2.75	1.58	2.570	1.62
8	3.00	1.50	2.450	1.55
9	3.22	1.43	2.340	1.48
10	3.49	1.35	2.208	1.39
11	3.62	1.32	2.148	1.36
12	4.00	1.20	1.960	1.24

**Table 2 polymers-15-01730-t002:** Artificial wound exudate composition according to Oates et al. [37]. The pH of the solution was adjusted to 7 using NaOH or HCl.

Compound	[Compound] (g·L^−1^)
MOPS	20.900
Sodium Chloride	6.025
Potassium Chloride	0.372
Urea	0.540
Creatinine	0.013
Glucose	0.324
Yeast extract	1.0
Peptone	3.0
Magnesium sulphate	0.017
Haemin	0.005
Potassium phosphate	0.109

**Table 3 polymers-15-01730-t003:** Maximum wavelength (λ_max_) values and molar extinction coefficient (ε) for AzoIz.Pip from pH 3 to 12 using Britton–Robinson buffered solutions.

pH	λ_max_ (nm)	Absorbance	ε (L·mol^−1^·cm^−1^)
3	557	1.19	5.79 × 10^3^
4	557	1.23	6.01 × 10^3^
5	557	1.23	5.97 × 10^3^
6	557	1.22	5.92 × 10^3^
7	557	1.21	5.90 × 10^3^
8	557	1.24	6.03 × 10^3^
9	544	1.32	6.43 × 10^3^
10	536	1.65	8.03 × 10^3^
11	536	1.74	8.46 × 10^3^
12	536	1.75	8.53 × 10^3^

**Table 4 polymers-15-01730-t004:** Color coordinates (L*–lightness, a*–yellowness/blueness, and b*–redness/greenness), K/S, ∆E, and UPF of the samples functionalized with AzoIz.Pip under the different conditions tested.

pH	T (°C)	L*	a*	b*	K/S sum	∆E ^#^	UPF	Sample
SK	-	76.1	−0.13	0.74	0.30	-	5	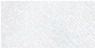
6	40	53.60	6.90	−24.99	0.58	33.99	8	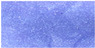
6	70	56.72	5.62	−21.38	0.52	29.06	9	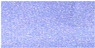
8	40	55.68	5.64	−22.05	0.54	30.23	8	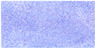
8	70	57.35	5.03	−20.72	0.51	28.05	7	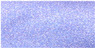

^#^ ∆E calculation using blank silk as control.

**Table 5 polymers-15-01730-t005:** K/S, ∆E, and UPF values of the dyed silk samples with different concentrations of AzoIz.Pip under pH 6 and 40 °C.

[AzoIz.Pip] μg·mL^−1^	K/S Sum	∆E ^#^	UPF
4	0.32	2.62	5
8	0.35	6.82	6
16	0.39	12.49	7
32	0.49	21.30	7
64	0.59	28.60	8
128	0.66	30.53	8

^#^ ∆E calculation using blank silk as control.

**Table 6 polymers-15-01730-t006:** Color coordinates (L*-lightness, a*-yellowness/blueness, and b*-redness/greenness), K/S, ∆E, and UPF of the samples functionalized with AzoIz.Pip and immersed in buffer solutions to assess the halochromic properties of the samples.

Buffer, pH	L*	a*	b*	K/S Sum	∆E ^#^	UPF	Sample
BR, 4	55.92	3.96	−16.73	0.54	1.50	7	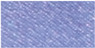
BR, 5	56.79	2.99	−15.71	0.53	1.80	7	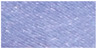
BR, 6	55.41	4.28	−17.45	0.55	1.18	7	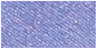
BR, 7	53.45	5.06	−18.09	0.59	2.25	8	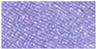
BR, 8	55.52	6.16	−18.59	0.54	3.19	8	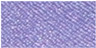
BR, 9	53.34	7.31	−20.02	0.58	4.19	8	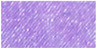
BR, 10	54.17	12.02	−20.22	0.55	9.16	9	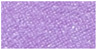
Artificial exudate, 7	55.33	3.86	−16.83	0.56	0.97	9	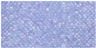
Artificial sweat, 6	54.23	4.06	−17.39	0.58	0.86	8	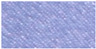
Artificial sweat, 8	54.13	4.43	−17.78	0.58	1.30	8	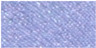

^#^ ∆E calculation using dyed silk under pH 6 at 40 °C as control.

**Table 7 polymers-15-01730-t007:** K/S, ∆E, and UPF values of the dyed silk samples with different concentrations of AzoIz.Pip under pH 6 and 40 °C and immersed in a Britton–Robinson solution at pH 10.

[AzoIz.Pip] μg·mL^−1^	K/S Sum	∆E ^#^	UPF
4	0.33	4.48	7
8	0.34	7.97	6
16	0.36	12.41	7
32	0.43	22.85	7
64	0.53	31.83	8
128	0.77	44.61	10

^#^ ∆E calculation using blank silk as control.

**Table 8 polymers-15-01730-t008:** Color coordinates (L*-lightness, a*-yellowness/blueness, and b*-redness/greenness), K/S, ∆E, and UPF of the samples functionalized with AzoIz.Pip and immersed in buffer solutions to assess the stability and leaching of the dye.

Buffer, Time	L*	a*	b*	K/S Sum	∆E ^#^	UPF	Sample
pH 5, 1 h	56.79	2.99	−15.71	0.53	4.27	7	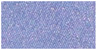
pH 5, 6 h	57.35	2.94	−15.4	0.52	4.91	7	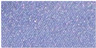
pH 5, 24 h	56.54	2.89	−15.21	0.53	4.37	7	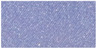
pH 7, 1 h	55.39	3.94	−17.35	0.55	2.29	9	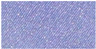
pH 7, 6 h	55.68	4.2	−17.41	0.54	2.59	8	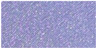
pH 7, 24 h	56.4	3.81	−16.56	0.53	3.48	8	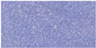
pH 8, 1 h	54.03	5.57	−19.23	0.57	2.46	9	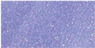
pH 8, 6 h	54.38	5.5	−18.45	0.57	2.23	8	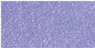
pH 8, 24 h	56.8	4.72	−16.63	0.52	3.96	7	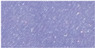
pH 9, 1 h	53.34	7.31	−20.02	0.58	4.19	7	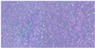
pH 9, 6 h	54.24	9.28	−19.9	0.55	6.02	8	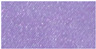
pH 9, 24 h	55.54	8.58	−18.19	0.53	4.42	7	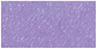
pH 10, 1 h	55.31	10.64	−19.25	0.53	7.38	8	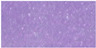
pH 10, 6 h	55.63	12.45	−19.17	0.52	8.68	7	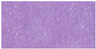
pH 10, 24 h	56.5	11.71	−17.18	0.50	9.16	8	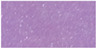

^#^ ∆E calculation using dyed silk under pH 6 at 40 °C as control.

**Table 9 polymers-15-01730-t009:** Color coordinates (L*-lightness, a*-yellowness/blueness, and b*-redness/greenness), K/S, ∆E and UPF of the samples.

	pH	L*	a*	b*	K/S Sum	∆E ^#^	UPF	Sample
Initialsample	6	53.60	6.90	−24.99	0.58	-	8	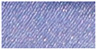
1st color change	10	54.57	11.8	−19.84	0.54	8.44	8	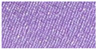
2nd color change	6	54.93	4.29	−18.12	0.56	1.87	8	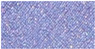
3rd color change	10	53.99	11.64	−20.12	0.55	8.28	8	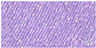
4th color change	6	55.17	3.69	−17.17	0.56	2.12	8	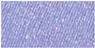

^#^ ∆E calculation using dyed silk under pH 6 at 40 °C as control.

**Table 10 polymers-15-01730-t010:** Results of the color fastness tests presented as ∆E variations and assessment of staining using the grey scale (green–not detected color change; blue–low color change detection; orange–strong color change detected).

	Acidic Sweat	Alkaline Sweat	Wound Exudate	Washing	Rubbing Wet	Rubbing Dry
∆E	0.86	1.30	0.97	14.48	-	-
Acetate	3–4	3–4	3–4	5	5	5
Cotton	4	4–5	4–5	5		
Polyamide 6,6	3–4	3–4	3–4	4		
Polyester	5	5	4–5	5		
Acrylic	4	4–5	5	5		
Wool	5	4	4	5		

**Table 11 polymers-15-01730-t011:** Component’s area extracted after C1s core level fitting by Gaussian function.

	C1	C2	C3	C4	Sample
Control	1.22 ± 0.06	0.68 ± 0.11	0.31 ± 0.08	0.51 ± 0.03	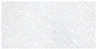
pH 6	1.34 ± 0.03	0.69 ± 0.04	0.62 ± 0.03	0.48 ± 0.02	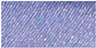
pH 10	1.38 ± 0.02	0.70 ± 0.03	0.64 ± 0.04	0.49 ± 0.02	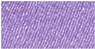

## Data Availability

Not applicable.

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
