# Peer review of "Halochromic Silk Fabric as a Reversible pH-Sensor Based on a Novel 2-Aminoimidazole Azo Dye"

_polymers, 2023, doi:10.3390/polym15071730_

Round 1

Reviewer 1 Report

The paper presents interesting results in which a new reversible pH sensor based on a novel 2-aminoimidazole azo dye is shown. The paper can be accepted for publication after minor revision. The following issues should be clarified:

It will be valuable to add information in the introduction about similar reversible pH sensors and their applications.

Why was the cytotoxicity evaluation conducted for the dyed silk extracts? I believe it is better to use the silk fabric and test the cellular behavior on its surface.

I suggest citing a paper where similar reversible pH sensors were developed: https://doi.org/10.3762/bjnano.10.233

https://doi.org/10.1039/C7AN00454K

Author Response

  1. It will be valuable to add information in the introduction about similar reversible pH sensors and their applications.

R1: Reversible pH sensors with a similar structure to the azoimidazole applied in this study are quite rare in the literature, once the synthesis of this type of molecules is commonly challenging due to the steric hindrance nearby the azo bond. More information about this topic can be found in the recent paper of our research group containing the synthesis of these molecules (https://doi.org/10.1039/D3CC00372H).

  1. Why was the cytotoxicity evaluation conducted for the dyed silk extracts? I believe it is better to use the silk fabric and test the cellular behavior on its surface.

R2: In this work, the indications of ISO 10993-5 to evaluate the cytotoxicity of medical devices were followed, in which extracts are recommended. The HaCaT cells adhere to the materials, and the cultivation of these cells on silk tissue is incompatible. That is the reason for using the extracts.

  1. I suggest citing a paper where similar reversible pH sensors were developed: https://doi.org/10.3762/bjnano.10.233, https://doi.org/10.1039/C7AN00454K

R3: The suggested references were added to the manuscript.

Reviewer 2 Report

This manuscript presents an interesting and thorough study on the functionalization of silk fabric with a novel class of 2-aminoimidazole azo dyes. The halochromic properties of the dye were assessed in aqueous solution and after silk functionalization. The results showed vibrant colors and attractive halochromic properties with a hypsochromic shift from blue to magenta in aqueous buffered solutions. Similarly, the functionalized silk showed a shift in wavelength of the maximum K/S value when pH increases.  

A few things to improve:

1.     The calculation of a, b and E relies on RGB value. Seems it’s unclear how the RGB value is determined?  

2.     It’s a nice and continent method to use color to determine pH. But the wound fluid and blood might have strong color staining. How would that affect the pH reading?

3.     Is the pH expected to be read by the naked eye or by any instrument?   Would the luminance of the environment make the color reading difficult?

4.     What’s the limit of detection of the sensor?  It can be calculated by DL = 3.3x σ / S 

Author Response

  1. The calculation of a, b and E relies on RGB value. Seems it’s unclear how the RGB value is determined?

R1: The RGB values were determined by measuring the reflectance of the textile samples using a Shimadzu UV-2600 spectrophotometer. The spectra were recorded from 280 to 800 nm with an interval of 1 nm using a D65 illuminant and a standard observer of 10 degree. Then, the reflectance data were used to determine the RGB values using the program UV-2401PC Color Analysis (color-shortcut). This information was added to the manuscript.

  1. It’s a nice and continent method to use color to determine pH. But the wound fluid and blood might have strong color staining. How would that affect the pH reading?

R2: In this work, only a solution that simulates the artificial exudate was used, which had a slightly yellow color. However, this did not influence the blue color of the dyed silk sample. In future works, the color change in the presence of fluids with stronger colors, such as the presence of blood, can be tested.

  1. Is the pH expected to be read by the naked eye or by any instrument? Would the luminance of the environment make the color reading difficult?

R3: The main purpose of this work is to read the pH by the naked eye in the presence of natural light (D65). Thus, additional studies would be needed to evaluate the visualization of color change in the presence of different illuminants and light intensities.

  1. What’s the limit of detection of the sensor? It can be calculated by DL = 3.3x σ / S

R4: In this case, the analyte is the pH of the solution and thus, the detection limit of the sensor was achieved with the pKa calculation of the molecule. The color change in the silk samples is verified when the pH is higher than 8.4 (pKa1 of the molecule).
